# Role of the European Pharmacist in the Implementation of the Latest WHO Guidelines for Malaria

**DOI:** 10.3390/pathogens12050729

**Published:** 2023-05-17

**Authors:** Anita Cohen

**Affiliations:** 1Faculty of Pharmacy, University of Aix-Marseille, IRD, AP-HM, SSA, VITROME, F-13005 Marseille, France; anita.cohen@univ-amu.fr; 2Assistance Publique-Hôpitaux de Marseille, Pharmaceutical Expertise and Clinical Research Unit, Pharmacy Department, APHM, F-13000 Marseille, France

**Keywords:** pharmacist, pharmaceutical advice, pharmaceutical analysis, public health promotion, WHO guidelines for malaria, personal protection measures, insecticide resistance, antimalarial chemoprophylaxis, continuous training

## Abstract

Following the publication a few months ago of the new WHO guidelines for malaria, the European pharmacist, even out of endemic areas, as a health care professional and advisor, has a major role to play in their effective implementation and in the interest of public health. The pharmacist is central in the health care system to ensure the correct application of these recommendations and to perform a decisive role in the prevention of malaria infections, both in the adapted pharmaceutical advice on personal protection against biting vector insects on the one hand, and in the pharmaceutical analysis and recommendations concerning antimalarial chemoprophylaxis prescriptions on the other hand. Physicians and pharmacist biologists, as well as hospital pharmacists, are also important for the analysis and treatment of effective malaria cases, particularly in the management of the diagnostic and therapeutic emergency that a *P. falciparum* infection represents.

## 1. Introduction

In June 2022, the World Health Organization (WHO) published guidance on new and updated recommendations [1], which were last updated on 14 March 2023 [2]. These guidelines encourage countries to tailor the recommendations to local disease settings for maximum impact. Globally, there were an estimated 247 million malaria cases in 2021 in 84 malaria-endemic countries (including the territory of French Guiana), an increase from 245 million in 2020, with most of this increase coming from countries in the WHO African Region [3]; since 2015, the WHO European Region has been free of malaria [3]. Nevertheless, in the era of “One Health” and alarming resistances to all anti-infective medicines [4,5] and the multiplication and facilitation of international transport which can greatly influence the spread of (vector born) diseases [6,7], the European pharmacist has a major role to play in dealing with this public health problem.

The main mission of the pharmacist is to contribute to primary care as defined by (i) prevention, screening, diagnosis, treatment, and follow-up of patients; (ii) dispensing and administration of medicines, products and medical devices, as well as pharmaceutical advice; (iii) guidance in the health care system and the medico–social sector; and (iv) health education. Whether in the pharmacy, in a medical laboratory, or in a hospital, the pharmacist is the most accessible health care professional specializing in medicines who is the link between the physician and the patient. Through their skills and knowledge, the pharmacist ensures optimal patient care in conjunction with general health promotion. Recently, the pharmacist’s place is still evolving in a regulatory and health context that tends to reorganize and transform the health system as in France [8], particularly by decompartmentalizing the careers of health professionals to create a “health workforce” at the service of patients and to structure the supply of care as well as possible [9]. As a highly trained health professional with adequate knowledge, by their central place in the health care system, the pharmacist has a major role in advising, patient education, and the management, control, and prevention of public health problems [10,11,12,13] such as malaria. Indeed, the pharmacist possesses the knowledge and skills required to ensure the safe and effective use of medicines [14] and is increasingly involved in health promotion and education [15,16,17,18].

In the application of the latest guidelines that call into question certain well-tried dogmas (e.g., insecticide no longer recommended for impregnation of clothing, chloroquine no longer recommended for chemoprevention), the pharmacist must be kept informed, regularly trained [13], and apply the principles of good health care professional–patient communication [19] to ensure their correct implementation. Their place in the health system and their role in patient education are essential in this context and they must transmit and promote clear messages to European travelers, mainly to reduce the impact of malaria at European and especially international levels. That is the reason why the pharmacist is an integral part of the health workforce recently founded by WHO to develop a strategy on human resources for health to 2030 [9].

## 2. Role in the Prevention of Malaria Infections

There is no question that since the beginning of the COVID-19 pandemic, pharmacists have been on the front lines of health care, and thus demonstrated and reinforced their role as a pivot in the system and in the prevention of infectious diseases [13,20]; with increasing activities and responsibilities, they are rapidly becoming the backbone of the health system and the first line of defense against this pandemic [21,22]. Within pharmacies, as a local health care professional, the pharmacist has a major role as a public health actor, as well as being a drug professional, in the prevention of infectious diseases, through their pharmaceutical advice and patient education activities. In the application of the latest WHO guidelines for malaria that call into question certain well-tried dogmas, the pharmacist must inform and aware patients of these important changes. The pharmacist therefore has a crucial role to play in raising awareness, avoiding misuse, and passing on information, bearing in mind that in Europe, the patients concerned are mainly travelers who will only occasionally apply these recommendations, but who are likely to contribute to the spread of many resistances [23]. 

### 2.1. Personal Protection against Biting Vector Insects

One of the biggest changes called for in the latest WHO guidelines for malaria concerns personal protection measures by no longer recommending the use of insecticide products for impregnating clothing. Initially introduced in 2019, in the guidelines for malaria vector control published by the WHO [24], the deployment of insecticide-treated clothing for malaria prevention is no longer recommended as an intervention with public health value; however, insecticide-treated clothing may be beneficial as an intervention to provide personal protection against malaria in specific population groups (refugees, military personnel, and others engaged in occupations that place them at high risk), on the basis of a Cochrane systematic review comparing insecticide-treated clothing versus placebo or untreated clothing in a randomized controlled clinical trial [25]. Indeed, this measure of insecticide-treated clothing was previously recommended for limited periods of time in cases of high exposure in the general population but was revealed to be ineffective and poses a risk of individual and environmental toxicity, which is now well documented [26,27]. This guideline is particularly important in the current context of the increasing insecticide resistance among malaria vectors because of the selection pressure from the increasing insecticide-based vector control. This resistance affects all major vector species and all classes of insecticides and has been of great concern for more than 10 years through the WHO Global plan for insecticide resistance management in malaria vectors in 2012 [28], then the Global vector control response for 2017–2030 in 2017 [29]. Furthermore, among new WHO recommendations, there is an initiative to stop the spread of *Anopheles stephensi*, a vector that has been expanding its range over the past decade in the WHO African Region and was found to be resistant to many of the insecticides used in public health, posing an added challenge to its control [3,30]. Among the key technical principles proposed by WHO to combat insecticide resistance are the following: (i) insecticides should be used carefully and thoughtfully to reduce unnecessary selection pressure; (ii) vector control programs should avoid using a single class of insecticide everywhere and for several consecutive years. Wherever possible, alternatives to pyrethroids should be used to maintain their effectiveness [29].

In this context, the role of the pharmacist is essential; as a local health care professional, they must transmit information to the public that insecticide-treated clothing is no longer recommended. It is thus their responsibility to change mentalities about a well-established dogma, and to prevent misuse by educating patients about this WHO directive. This is more important as the resistance of vector agents to insecticides is widespread and increasing. Indeed, insecticides are still available even if these products, represented by pyrethroids (mainly permethrin in Europe), now have an unfavorable benefit–risk balance for impregnating clothing in the general population, whereas they are always used for the impregnation of recommended mosquito nets. The pharmacist also has a role to play in monitoring and managing this resistance [31] and enforcing the regulation for this category of products which are classified as biocidal pesticides [32]. The pharmacist must therefore promote the use of recommended personal protection measures, including (i) to protect against insect bites, including with repellents, especially on uncovered areas; (ii) to sleep under a mosquito net at night, preferably one that has been impregnated with insecticide (properly installed and ensuring the integrity of the mesh); and (iii) to wear light, loose-fitting, covering clothing (long sleeves, trousers, and closed shoes) [33]. Moreover, the pharmacist must ensure proper use of these individual measures, especially concerning insecticide products.

### 2.2. Antimalarial Chemoprophylaxis

Another dogma undermined by the latest WHO guidelines for malaria concerns the chemoprophylaxis, since chloroquine is no longer recommended for malaria chemoprophylaxis [1]. Nevertheless, this change was largely heralded by the global spread of chloroquine resistance in recent decades and the gradual cessation of marketing of several chloroquine-based medicines in recent years (in the France cessation of marketing of the chloroquine-proguanil combination since July 2020, the pediatric form of chloroquine syrup since July 2021, and finally, the cessation of manufacturing of chloroquine tablets since September 2022). The pharmacist must therefore, through their pharmaceutical analysis of the prescriptions, in front of any possible prescription of chloroquine, inquire about the indication and contact the prescribing physician without delay to modify the prescription in favor of a recommended chemoprophylaxis (atovaquone-proguanil or doxycycline in first intention, mefloquine in last intention). Again, the pharmacist has to ensure the proper use of the recommended atovaquone-proguanil and doxycycline and their respective conditions of administration through comprehensive information and effective patient education. These two antimalarial drugs have a high and comparable efficacy. There is also mefloquine with comparable efficacy, but potential serious adverse reactions, too (neuropsychiatric disorders from headaches, unexplained sadness, nightmares to acute anxiety, depressive syndrome, agitation, mental confusion, and suicidal ideation), mainly found several years after its marketing authorization by pharmacovigilance [34,35]. It is the pharmacist’s duty to echo the physician’s recommendations regarding this potential risk of neuropsychiatric disorders during mefloquine treatment, a risk that persists for up to several months after stopping the drug. Patients treated with mefloquine chemoprophylaxis for malaria should be informed that if such adverse effects occur, they should immediately discontinue treatment and consult a physician to change the chemoprophylaxis. In these conditions, mefloquine should only be considered as a last scheme in adults, except for long-term travelers at higher risk for whom mefloquine should be considered as the first-choice chemoprophylaxis [36]. Nevertheless, mefloquine presents a significant advantage of weekly administration because of its long half-life, which may improve observance and compliance if individual tolerance has been previously assessed and tested at least 10 days before the date of entry into a malaria-endemic area.

### 2.3. Specific Pharmaceutical Advice

#### 2.3.1. Use of Herbal Teas or Dietary Supplements with *Artemisia annua*

Based on the above, the pharmacist has a major role to play in advising and recommending malaria prevention measures. This is even more important to prevent the risk of non-recommended measures, such as the misuse of herbal teas or dietary supplements with *Artemisia annua*, the plant from which artemisinin and some derivatives [37], the first-line antimalarial drugs class [38], are extracted. The currently recommended curative antimalarial treatment is mainly based on parenteral artesunate (as an emergency treatment for severe *P. falciparum* malaria) and oral artemisinin-based combination therapies (ACT) [1]. ACTs are generally highly effective and well tolerated. This has contributed substantially to reductions in global morbidity and mortality from malaria during the last decades. Unfortunately, resistance to artemisinin has arisen since 2009 in *P. falciparum* in Southeast Asia [39], and more recently, a de novo artemisinin partial resistance has emerged in the WHO African Region [3], which threatens these gains. In this worrying context, the use of artemisinin and its derivatives must be perfectly organized, justified, and controlled to limit selection pressure, preserve their effectiveness, and slow down the progression of these resistances.

However, the misuse of artemisinin-based dietary supplements or herbal medicine based on dried *Artemisia* plants as antimalarial prophylaxis is increasing among travelers and raises the risk of severe malaria complications that can lead to death [40]. Moreover, following an extensive review of the evidence on the efficacy of non-pharmaceutical forms of *Artemisia* conducted in 2019, WHO issued a policy statement in which it does not justify the promotion of *Artemisia* plant material or its use in any form for the prevention or treatment of malaria [41]. In these conditions, it is the responsibility of the pharmacist as a health promoter and community health professional to inform, educate, and explain to patients the risks associated with this misuse and to advise and dispense only products that meet the WHO recommendations. 

#### 2.3.2. Special Conditions

Another important element that must be considered by the pharmacist during their pharmaceutical analysis and travel advice is the planned conditions of the trip. Indeed, a stay in an all-inclusive residence will not require the same protective measures as a trek under the stars. Similarly, protection measures will not be the same if some travelers belong to special populations including babies, children, pregnant women, nursing mothers, and also the elderly and people with comorbidities or allergies. Risk assessment conducted by health professionals, including the community pharmacist in the front line, must therefore be individual and based on a detailed study of travel conditions (length of stay, habitat, areas visited, activities, etc.) and the profile of each traveler (age, weight, pregnancy, comorbidities, treatments, etc.) [33]. Indeed, through their skills, their knowledge of the patient, and of the authorized and available medicines, the pharmacist must constantly adapt their advice and recommendations in the context of health care in general [13]. As an example, in terms of personal vector protection, specific products exist for pregnant women and young children, such as minimum concentration skin repellents [42]. Thus, the pharmacist must be attentive to the patient’s history and/or pathologies and/or allergies and/or chronic treatments to minimize any risk of possible drug interaction or contra-indication. The pharmacist must also be continuously trained on the different measures of protection, ongoing developments [43], such as the use and characteristics of the malaria vaccine RTS,S/AS01 [1,44] or the new types of insecticide-treated nets (pyrethroid and piperonyl butoxide treated nets) [1,23,45], even if these tools do not concern travelers and are therefore not available in Europe to date.

## 3. Role in the Management of Effective Malaria Infections

Although the WHO European Region has been free of malaria since 2015 [3], all countries in this area should maintain vigilance to rapidly detect imported malaria cases. Beyond their role as a local health professional who delivers expert pharmaceutical advice adapted to the patient’s profile, the pharmacist also has a crucial role in the management of the patient suffering from malaria, whether in the immediate referral to a medical consultation of any patient suffering from fever and having traveled in an endemic zone, or in the biological analysis for the diagnosis and characterization of the infection or in its treatment. Indeed, *P. falciparum* infection constitutes a diagnostic and therapeutic emergency because of the risk of life-threatening complications, and the pharmacist, according to their area of expertise and competence, must be able to detect, identify, and treat this emergency.

### 3.1. Detection of Potential Malaria Cases

As previously mentioned, pharmacists, as local health professionals, must be aware of patients with fever and/or gastro-intestinal symptoms if they have traveled in an endemic zone, and thus suspect malaria as a potentially severe disease, and advise an immediate referral to a medical consultation. In this framework, the pharmacist must develop the principles of good practices of communication in health care and apply them [18]. Using the Calgary Cambridge Observation Guide [46] and its revised content version [47], the health care professional could learn a method and some principles to apply during the patient interview in real-life practice, which may be helpful in the implementation of their competences for the health care professional’s missions of health monitoring, patient education, and public health promotion. 

### 3.2. Diagnostic of Potential Malaria Infections

While the WHO European Region has been free of malaria since 2015, there were approximately 5000 cases of imported malaria observed each year in metropolitan France, which remains the most affected country in continental Europe by malaria outside of the endemic zone [48]. These imported malaria cases might occur anywhere, at any time, and must be detected as soon as possible. Even if rapid diagnostic tests (RDTs), used in countries with endemic malaria, are available for diagnosing imported malaria in non-endemic, high income countries, a recent review concluded that it cannot be recommended that RDTs replace the gold standard of microscopy [49], but they can contribute to the establishment of the rapid diagnosis. European physicians and pharmacist biologists must therefore be vigilant and efficient in their analysis, diagnosis [50], and management of effective malaria cases, particularly in the management of the diagnostic emergency that a *P. falciparum* infection represents. Currently, the techniques recommended for the diagnosis of malaria in France combine a high-sensitivity method (thick blood smear, QBC test, or rapid molecular biology testing) and a thin blood film (parasitemia evaluation and species identification via characterization of *Plasmodium* morphological stages) [51]. Nevertheless, in practice, a thin blood film and an RDT are used as an alternative when this algorithm cannot be performed. If RDTs are used, it is required that they detect the *P. falciparum*-specific HRP2 antigen [51]. If the initial result is equivocal or negative, then testing should be repeated 12 to 24 h later [51]. PCR control testing can also be performed in a reference laboratory (presence of multiple species or infection notably related to low parasitemia) [51]. Through their knowledge and skills, the pharmacist biologist masters all these techniques and optimizes their implementation in terms of time and ever more efficient equipment to enable the rapid identification of cases of *P. falciparum* requiring emergency treatment.

### 3.3. Treatment of Effective Malaria Infections

As previously mentioned, a *P. falciparum* infection constitutes a diagnostic and therapeutic emergency because of the risk of life-threatening complications. According to the WHO Guidelines [1], the treatment to be administered without delay is based on artesunate, a hemisynthetic derivative of artemisinin which does not have a marketing authorization in Europe. However, an injectable form is manufactured by Guilin Pharmaceuticals which has a marketing authorization in China. To date, ACE Pharmaceuticals imports this specialty, recontrols it, and then distributes it in certain European countries for compassionate use as Artesunate 60 mg powder and solvent for injectable solution. This use is subject to a close monitoring procedure by the national regulatory authorities [52], particularly in terms of pharmacovigilance.

As Artesunate 60 mg powder and solvent for injectable solution is used in clinical emergencies, this product is for hospital use only and must always be on reserve within the health care institution. Thus, the hospital pharmacist must build up and manage their emergency stock, but also ensure all the stages of the dispensing of this product in compassionate access (verification of access criteria, dispensing, management, and administrative traceability in connection with the regulatory authorities). In adults and children over 20 kg, the recommended IV dosage [1] is 2.4 mg/kg/12 h for 24 h, then every 24 h for up to 7 days (maximum 9 doses). In children under 20 kg, the dosing regimen remains the same, but the dose is 3 mg/kg. This treatment is thus continued in all ventilated intubated patients or until an oral relay can be considered after the initial severity criteria have disappeared. Oral relay is intended to avoid late recurrences and is mandatory. The antimalarial drugs to be used are preferably fixed combinations including an artemisinin derivative [1]. These artemisinin-based combination therapies are dispensed by the pharmacist in hospitals and pharmacies.

## 4. Conclusions

In the current context of health crisis and increasing resistance globally, infectious diseases, including malaria, are more than ever a public health problem and an absolute priority. In a health system in full evolution, to decompartmentalize the careers of health professionals and optimize the supply of care, the pharmacist has a pivotal and growing place in health promotion and the implementation of public health recommendations. This is particularly true in the case of malaria, where the latest WHO recommendations overturn certain established dogmas. In Europe, an area free of malaria for several years, the pharmacist, depending on their field of expertise in the pharmacy, medical laboratory, or hospital, has a key role in the proper implementation of these recommendations and in health promotion. Furthermore, in their mission of prevention, the pharmacist can participate in the research and development of new diagnostic tools and treatments through the preclinical development and/or the implementation of clinical trials on innovative medical devices, molecules, or combinations in the diagnostic and treatment of malaria. Thus, there are 15 registered clinical trials on malaria in France in the ClinicalTrials.gov database accessed on 13 May 2023, of which 10 have recruited patients in the last 10 years [53]. 

In Europe, but also in endemic areas, intensive cooperation is needed between pharmacists, physicians, other health professionals, and travel medicine stakeholders to ensure the proper implementation of the public health guidelines and to achieve the health workforce desired by the WHO to accelerate progress toward universal health coverage and sustainable development goals.

## Data Availability

Not applicable.

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
