# Peer review of "Role of the European Pharmacist in the Implementation of the Latest WHO Guidelines for Malaria"

_pathogens, 2023, doi:10.3390/pathogens12050729_

Round 1
Reviewer 1 Report
The subject of this paper is relevant and potentially educative. The specific aim of the paper, however, is not clear. If the aim is to educate the public, policy makers and health professionals on the role of the pharmacist in the health care system and to motivate the pharmacist, the structure or style of writing has to be revised. The current style of the paper does not convey this relevance and /or education. The author focused on recommendations relating to prevention and effective treatment of malaria as specified in the WHO Guidelines for malaria and defined a role for the pharmacist in the recommendations. The author mainly “explained” the recommendations and in doing so mentioned the “role” of the pharmacist. The write up is thus muddled. A bit of the explicit roles of the pharmacist can be seen in the introduction (lines 35 and 36) but generally, the over-elaborated explanations of the recommendations drowned the role of the pharmacist.
The pharmacist is a highly trained health professional with adequate knowledge and skills set and has a PLACE in the health care system at all levels. It is that knowledge and skills set the health system draw on in the implementation of any health policy. It is important for a paper like this to draw attention to the pharmacist in the provision of health care in general and in particular malaria.
An alternative style for consideration by the author:
In the introduction, indicate the resourcefulness of the pharmacist in the health care system because of their knowledge and skills set and mention the roles these have equipped them to play generally. The body should mention individual roles in the context of the recommendations specified in the WHO Guidelines for malaria. Major roles such as advising, patient education, and the management and prevention of malaria will be sub-headings.
Author Response
Dear Reviewer,
First, I would like to thank you for your review report.
I am writing to you regarding the submission of the revised version of the communication titled: “Role of the European pharmacist in the implementation of the latest WHO guidelines for malaria”.
All revisions suggested have been considered:
As demanded, the structure of writing has been revised to clarify the specific aim of this paper which is to inform on the place of the pharmacist in the health care system and to identify his specific role at each stage of the patient’s care in the context of the implementation of the latest WHO Guidelines for Malaria.
Thus, the place of the pharmacist and his general missions have been detailed in the introduction as suggested, but also through the text in a more specific way at each stage of the patient’s care in the context of the recommendations from the latest WHO Guidelines for Malaria (prevention, detection, diagnostic and treatment of malaria infections). Furthermore, the bibliography references have been enriched to better illustrate the place of the pharmacist in the public health system and his role in health promotion and patient education.
All these elements were taken up in the conclusions with an opening of the subject on the interprofessional cooperation required in a global public health objective.
Moreover, a few errors have also been corrected.
Waiting for your appreciation,
Sincerely yours,
Anita COHEN.

Reviewer 2 Report
The article is a short communication accompanied by some personal comments about the role of the European pharmacist in the implementation of the latest WHO Guidelines for malaria.
Given that the article is well written, even if it has only 7 but it is well documented (43 references) and the scientific relevance is not clear to me, I agree with the publication after minor changes. 1. To complete a few conclusions 2. To give up the trade name MALACEF. Is it enough Artesunate 60 mg, powder and solvent for... 3. In italics for all ... P. malariae and Artemisia annua . 4. page 5, line 203, to be corrected morpholigical in morphological.
Author Response
Dear Reviewer,
First, I would like to thank you for your review report.
I am writing to you regarding the submission of the revised version of the communication titled: “Role of the European pharmacist in the implementation of the latest WHO guidelines for malaria”.
All revisions suggested have been considered:
- As demanded, the conclusions have been completed to clarify the specific aim of this paper: The place of the pharmacist and his specific missions have been identified generally and specifically in the context of the recommendations specified by the WHO Guidelines for Malaria, with an opening of the subject on the interprofessional cooperation required in a global public health objective.
- The trade name MALACEF® has been deleted as demanded, in favor of the mention “Artesunate 60 mg, powder and solvent for injectable solution”.
- Words in foreign language such as Plasmodium, Artemisia annua… appear from now in italics throughout the text.
- The term “morphological” has been corrected (line 244 of the revised version).
Waiting for your appreciation,
Sincerely yours,
Anita COHEN.

Round 2
Reviewer 1 Report
The authors have addressed my concerns and I have no further comments.